# Austrian Veterinarians’ Attitudes to Euthanasia in Equine Practice

**DOI:** 10.3390/ani9020044

**Published:** 2019-01-30

**Authors:** Svenja Springer, Florien Jenner, Alexander Tichy, Herwig Grimm

**Affiliations:** 1Unit of Ethics and Human-Animal-Studies, Messerli Research Institute, University of Veterinary Medicine, Vienna, Medical University of Vienna, University of Vienna, 1210 Vienna, Austria; Herwig.Grimm@vetmeduni.ac.at; 2University Equine Hospital, University of Veterinary Medicine, Vienna, 1210 Vienna, Austria; Florien.Jenner@vetmeduni.ac.at; 3Department of Biomedical Sciences, University of Veterinary Medicine, Vienna, 1210 Vienna, Austria; Alexander.Tichy@vetmeduni.ac.at

**Keywords:** euthanasia, equine veterinary medicine, questionnaire-based survey, veterinary medical ethics

## Abstract

**Simple Summary:**

Euthanasia of companion animals is a challenging responsibility in the veterinary profession. Convenience euthanasia, over-treatment of animals, and financial limitations often present challenging situations in veterinary practice. Only a few empirical investigations have been published which concentrate on the horse owner’s perspective on euthanasia in equine practice. Data findings on veterinarians’ attitudes toward euthanasia in equine medicine are even scarcer. To this end, an anonymous questionnaire-based survey of Austrian equine veterinarians’ examines attitudes to the euthanasia of equine patients in a range of scenarios; to identify factors which may influence decisions on the ending of a horse’s life. The study showed that veterinarians consider contextual and relational factors in their decision-making. They are aware of owners’ emotional bonds with their horses and financial background, however, requests for convenience euthanasia are typically rejected. Although some significant differences between the tested variables, e.g., gender and working experience emerged, the attitudes of the veterinarians were shown to be largely shared. We conclude that veterinarians are aware of the multiple factors that influence their decision-making and gave indications as to the weight of animal- and owner-related factors in the handling of euthanasia in equine practice.

**Abstract:**

Euthanasia of companion animals is a challenging responsibility in the veterinary profession since veterinarians have to consider not only medical, but also legal, economic, emotional, social, and ethical factors in decision-making. To this end; an anonymous questionnaire-based survey of Austrian equine veterinarians examines the attitudes to the euthanasia of equine patients in a range of scenarios; to identify factors which may influence decisions on the ending of a horse’s life. This paper describes the distributions of demographic and attitude variables. Mann-Whitney U tests were used to test the associations of gender, work experience, and equine workload with attitudes in relation to euthanasia statements and case scenarios. In total, 64 respondents (response rate = 23.4%) completed the questionnaire. The study showed that veterinarians consider contextual and relational factors in their decision-making. They are aware of owners’ emotional bonds with their horses and financial background, however, requests for convenience euthanasia are typically rejected. Although some significant differences between the tested variables emerged, the attitudes of the veterinarians were shown to be largely shared. In conclusion, veterinarians are aware of the multiple factors that influence their decision-making and gave indications as to the weight of animal- and owner-related factors in the handling of euthanasia.

## 1. Introduction

Euthanasia of companion animals is a challenging responsibility in the veterinary profession and has received increasing attention in recent years [1,2,3,4,5,6,7,8,9,10,11,12,13,14,15,16,17,18,19]. Besides the medical issues, legal, economic, emotional, and social factors, as well as ethical concerns, are involved and influence the decision-making processes. Against this background, veterinarians need to consider not only the presumed interests of the animal patient but also the owner’s views and wishes [12,15]. These potentially conflicting responsibilities lead to challenging complexities in veterinary care. The treatment of equine patients is often associated with significant costs which owners may be unwilling or unable to pay. This means that the issue of euthanasia performed for financial reasons where curative or palliative therapies are available is of particular interest in equine practice. In some situations, owners might refuse to go ahead with medically indicated euthanasia of a horse because, due to their strong emotional bonds to the horse, they cannot bear to part from it. Thus, convenience euthanasia, the over-treatment of suffering animals and the influence of financial considerations on treatment decisions often create ethical dilemmas in veterinary practice [10,11,18,20].

A growing number of scientific investigations on euthanasia have been published in recent years, but these are mainly in the field of small animal practice [9,10,11,14,18]. Although a few empirical investigations concentrating on the horse owner’s perspective on euthanasia do exist [21,22,23,24], there is a lack of data focusing on the issue of veterinarians’ attitudes to equine euthanasia. Therefore, the rationale of the present study was to gather data on the attitudes of Austrian equine veterinarians toward euthanasia. The objective was to investigate these attitudes in various scenarios and to identify factors related to the patient, as well as to the client, which may influence practitioners’ decision-making when they are faced with the possibility that a horse’s life needs to be ended. A range of variables, including gender, work experience, and working time spent with horses was analyzed, and their significance for the veterinarian’s attitude to euthanasia assessed. This approach was designed to shed light on the complex issue of euthanasia from the veterinarians’ perspective, thereby testing the empirical hypothesis that the decision-making processes involved are highly contextual and multifactorial.

## 2. Materials and Methods 

### 2.1. Questionnaire Design

The online questionnaire, entitled “Euthanasia in equine practice”, was developed with reference to an already conducted Austrian survey on euthanasia in small animal practice [14]. In total, it comprised 56 questions arranged in five question categories: general/demographic (n = 12, Table 1); medical/technical (n = 5); (dis-)agreement with normative and descriptive statements (n = 25); case scenarios (n = 10); and open-ended questions (n = 4). Against the statements, see Table 2, and case scenarios, see Table 3, respondents were asked to rate their (dis-)agreement on a scale from 1 to 9, thereby providing information on their attitudes in each case. At the end of the questionnaire, respondents had the opportunity to write comments. The final version of the questionnaire was prepared with the help of members of the Working Group on Ethics at the Equine Hospital. The group is made up of veterinarians and an ethicist with posts at the Equine Hospital of the University of Veterinary Medicine who work together on ethical issues arising in the daily working life of practicing veterinarians, especially in the context of euthanasia [25]. The questionnaire was piloted with six veterinarians from the working group, and their comments and suggestions were incorporated into a final version before the administration of the survey. The survey was conducted in German, see Appendix A, as an online survey. A description of the aims of the study was supplied in addition to the survey questions. An English translation of the survey is presented in Table 1, Table 2 and Table 3, see Appendix A. 

Ethical approval was obtained through the Ethics Committee of the Medical University of Vienna.

#### 2.1.1. Design of Statements

The questionnaire contained 25 statements. These were thematically classified into the following four categories: (1) general statements on euthanasia (n = 6; AI5, AI8, AII4, AII8, AIII3, AIII7); (2) statements related to patients (n = 6; AI1, AI2, AII3, AII6, AIII2, AIII5); (3) client-centered statements (n = 10, AI3, AI4, AI6, AI7, AII2, AII5, AII7, AIII1, AIII4, AIII6); and (4) statements including technical and medical factors (n = 3; AI9, AI10, AII1). Respondents were asked to indicate their attitude towards each statement using a scale from 1 (“I do not agree at all”) to 9 (“I completely agree”). The median response for the level of agreement and its interquartile range (IQR) were calculated and reported in Table 2.

#### 2.1.2. Design of Case Scenarios

To gain deeper insight into the attitudes of veterinarians to euthanasia under specific conditions, participants were presented with several contextualized case scenarios. As with the statements, the scenarios address different thematic categories, covering reasons for refusing euthanasia (a strong emotional bond (F1), horse neglect (F8), euthanasia technique (F10), or “convenience euthanasia” (F2, F3, F4, F5, F7)) and the decision-making process for or against euthanasia (F6, F9).

### 2.2. Selection of the Survey Population

All referring veterinarians at the University Equine Hospital and members of the Austrian Equine Veterinary Association (VÖP) were invited to participate in the survey. In total, 273 veterinarians received an email with a link to the online survey, to which they had access between 24 May and 3 July 2016. The anonymity of all respondents was assured. No additional measures, such as follow-up telephone calls to individual veterinarians, were taken.

### 2.3. Data Analysis

Data from the questionnaire were collected in Google Forms and exported to Microsoft Excel data sheets. Descriptive statistics including means, standard deviation (SD), median, and IQR were created in Microsoft Office Excel 2011 (Microsoft Corp., Redmond, WA, USA). Further analyses were performed using the statistical software package SPSS version 24.0 in order to test the statistical significance of statements and case scenarios in relation to different variables, e.g., age, gender, the average duration of professional activity, and working time with horses (IBM Corp. Released 2016. IBM SPSS Statistics for Windows, Version 24.0. Armonk, NY: IBM Corp). A non-parametric *t*-test (Mann-Whitney U Test) was utilized in a univariable analysis, with distinctions being considered statistically significant if *p* < 0.05. This paper will focus on the analyses of gained data of demographic and general variables, 25 statements, as well as 10 case scenarios, of the survey. 

## 3. Results

Out of the 273 veterinarians contacted, 64 completed the questionnaire sufficiently fully and were included in the analysis (response rate = 23.4%). In Table 1, the demographic and general variables are summarized for the population of respondents as a whole and then listed separately for female and male veterinarians. 

Among the 28 respondents stating that they devoted 60% or less of their working time to horses, the proportion of those who treated small animals as well as horses was 73.1% (n = 19) for women and 26.9% (n = 7) for men. Furthermore, 28.6% (n = 6) of male and 4.7% (n = 2) of female veterinarians worked with farm animals in addition to horses. Additionally, the questionnaire asked how often euthanasia was performed per month. In total, 58 veterinarians gave answers that could be analyzed. The maximum value was ten per month, and the median value was one per month (IQR (0.5;1.5)). The veterinarians were also asked how many times they are asked to euthanize a horse against their professional opinion or advice per year. Here, 57 answers could be analyzed. The maximum value was 30 requests per year, and the median value was one request per year (IQR (0.0;1.5)). 

### 3.1. Statements 

Table 2 presents 25 normative and descriptive statements together with corresponding calculated median values and IQRs. Statistical analysis resulted in significant differences in median agreement scores based on the respondent’s gender or the percentage of equine work where five statements were concerned.

The statement that, in general, it would be difficult to euthanize against a veterinarian’s own conviction resulted in a significant difference between male and female veterinarians (*p* < 0.05), although median score (9.0 versus 9.0) and mean score (8.4 versus 9.0) do not indicate a high difference. Nevertheless, all 40 women chose the option nine on scales compared to their male colleagues. Among 20 men, three male veterinarians each chose option 1, 6, or 8 once. This could be a possible explanation for the significant difference. Further, male veterinarians indicated stronger agreement than female colleagues (median score 9.0 versus 7.0; mean score 8.14 versus 6.86) with the statement that carefully considered euthanasia as a positive part of their practice as veterinarians (*p* < 0.05). The statement that treating the owner with compassion is a central part of euthanasia resulted in a significant difference between veterinarians with a working experience ≤15 years and a working experience over 15 years (*p* < 0.05), even though the median score (9.0 versus 9.0), as well as the mean score (8.89 versus 8.29), do not indicate a high difference. 

Working time spent with horses in practice resulted in a significant difference in the case of two statements. Veterinarians devoting ≤60% of their working time with horses indicated a higher level of agreement with the statement that it is easier for them to deal with euthanasia if the horse is old (*p* < 0.05) than those devoting >60% of their time to horse care (median score 8.0 versus 7.0; mean score 7.04 versus 5.49). Additionally, veterinarians devoting ≤60% of their working time to horses indicated a higher level of agreement if they know that they thoroughly informed the horse owner about euthanasia (*p* < 0.05) than those devoting >60% of their time to horse care (median score 9.0 versus 8.5; mean score 8.38 versus 7.53).

### 3.2. Case Scenarios

Table 3 presents veterinarians’ level of agreement with the endorsement of euthanasia in ten scenarios together with the corresponding calculated median values and IQRs. Statistical analysis resulted in significant differences in the median agreement score based on respondents’ gender, work experience, or the percentage of equine work where three statements were concerned. 

Veterinarians spending more than 60% of their working time with horses are more likely to refuse to euthanize the healthy survivor of two geriatric horse friends even though the late owner’s will stipulates that the second horse is to be euthanized if the other dies (*p* < 0.05) compared to those spending less than 60% of their time with horses (median score 1.0 versus 1.5; mean score 1.83 versus 3.29). Veterinarians with less working experience are more likely than those with more experience (median score 1.0 versus 1.0; mean score 1.0 versus 1.91) to refuse to euthanize a stallion with a very good prognosis for purely financial reasons (*p* < 0.01). 

## 4. Discussion

In general, the consideration of medical and technical aspects seems to be central for equine veterinarians. In particular, the importance of a technically flawless procedure attracted overwhelming agreement (AI10). This result resembles those of Endenburg and colleagues [22] and Brackenridge and colleagues [24], who identified that well-performed euthanasia reduces stress, not only for the animal patient but also for the owner, who is likely to be present during the procedure [22,24]. Hence, training institutions are well advised to provide effective training and education, imparting the necessary knowledge and skills for euthanasia, especially from the medical perspective to reduce stress for their graduates. In addition, veterinarians can rely on the steadily increasing number of published guidelines, as these provide detailed information about technical matters and requirements [26,27]. Interestingly, it was not only technical components that mitigated the stress involved in dealing with euthanasia. Characteristics of the horse such as advanced age, a fulfilled life, and short life expectancy also help veterinarians to deal with euthanasia more easily and calmly (AI2, AII3, and AIII5). The veterinarians also strongly agreed that respectful treatment of the dead animal is a crucial component of euthanasia (AII6). Alongside traditional animal removal, demand for alternative animal burials is increasing in the equine sector, and this provides not only animal owners but also veterinarians, with opportunities to guarantee respectful treatment of the dead animal.

A further objective of the study was to obtain an improved understanding of veterinarian attitudes to owners, and to some specific owner-related issues. It emerged that veterinarians are aware of their information-giving and advisory role, as there was wide agreement that providing sufficient information to owners (AI3), ensuring that they possess a clear understanding of the need for euthanasia (AI4), and treating them in a compassionate way (AII7), are crucial elements in the veterinarian’s handling of euthanasia. This is in keeping with owners’ attitudes: owners appreciate a clear explanation of the procedure, and the provision of insufficient information can lead to owner dissatisfaction [22].

Veterinarians consider not only medical but also social and economic factors, to be of great importance (AI5). The emotional bond between the owner and the horse has an especially marked impact on decision-making processes. The survey respondents indicated that they would not continue to try to convince the owner of a severely laminitic mare of the necessity to euthanize once she/he had refused euthanasia despite in-depth explanations (F1). The most likely interpretation of this result is that veterinarians consider close emotional bonds to be of high importance and may accept that the animal will continue to suffer from its disease, at least to a certain extent, if the owner cannot part from the horse. This interpretation is supported by the respondents’ ambivalent responses to the statements that veterinarians can deal with a horse’s suffering if they know that they have done their best for the horse’s well-being (AI1) or have carefully informed the owner who does not agree to euthanasia of the animal (AII2). The case example and statements illustrate the issue of over-treatment—a problem which has also been described in the field of small animal medicine and leads to challenging situations and ethical dilemmas for practicing veterinarians [4,6,10]. When equine patients are suffering from incurable diseases and owners decide against medically indicated euthanasia, conflicting responsibilities create an ethical dilemma and role conflict for the treating veterinarian [4,6,10]. As a possible solution, veterinarians can strengthen their advisory position and inform the owner in greater detail about the animal’s suffering, explaining the reasons why euthanasia is the best possible solution [15]. Austrian veterinarians can, in addition, notify the official veterinarian, who is recruited by the government to perform authorized duties on its behalf, in cases where the condition of the animal is no longer bearable for animal welfare reasons. In the case of the neglected horse, veterinarians indicated high levels of agreement that they would inform the official veterinarian. Hence, it can be assumed that this may also be the ultima ratio decision in cases of over-treatment of a suffering animal.

Male veterinarians indicated a higher level of agreement with the statement that carefully considered euthanasia is a positive part of their practice as veterinarians (AIII7). This difference echoes a German study of social competencies relating to the euthanasia of dogs in which female veterinarians reacted more sensitively to owners’ emotions and were more willing to reflect on situations relating to euthanasia than their male colleagues [28]. A possible reason for the difference is that women are more concerned than men about the death of the euthanized animal on the one hand [28]. While, on the other hand, female respondents in our study were younger and less experienced than males and this may also affect their training and attitudes towards euthanasia.

Although veterinarians know they have limited influence on owners’ decisions, they agreed that it would be very difficult for them to euthanize against their own convictions (AII4). Owner-related statements indicating, for example, the possibility that the owner will neglect the horse (AIII1) or that the veterinarian will lose the client did not seem to influence veterinarians’ decisions on euthanasia (AIII6). Indeed, they were associated with a strong refusal to euthanize the patient in the absence of a medical indication. Results from the case scenarios highlighting the issue of convenience euthanasia provided further evidence for this veterinarian attitude. Four out of five convenience euthanasia requests resulted in a strong refusal to euthanize the horse despite the external pressure applied by the owner. Neither changed living circumstances, the last will of the owner, nor financial reasons led to an agreement to euthanize the horse. This accords with results in the field of small animal medicine [9,14]. Austrian veterinarians working with small animals have also confirmed that rationales referring to time and financial problems cannot justify animal euthanasia and that the principle of the protection of life has priority over the owner’s interests [14]. 

The Austrian legal system may explain the results of both studies, as it makes it illegal to kill an animal without justification (Austrian Animal Welfare Act 2004) [29]. In addition, it is clear that alternative solutions, such as finding a new home for the animal or offering payment in installments, can offer alternative options. However, it needs to be borne in mind that in equine medicine owners are more often confronted with high therapy costs and that finding a new home for a horse is a much greater challenge than finding the same for a dog or cat. As a result, the differing financial backgrounds of owners demand high flexibility in the planning and provision of medical services. With the increasing number of cost-intensive diagnostic and therapeutic options, financial considerations are becoming more prominent in today´s veterinary practice, and in both equine and small animal practice they cannot be uncoupled from the euthanasia issue [15,30].

Although the veterinarians took a very clear stance on the cases and statements already mentioned, some situations resulted in ambivalent attitudes. Respondents were ambivalent, for example, about the case of the foal with symptoms of severe colic (F5), resulting in a median value of 5. This situation would require a rapid decision from the veterinarian, who (we can imagine) will be distressed at the poor general condition of the foal, on the one hand, and the owner’s denial of the necessary therapy, on the other. This case appears to be very controversial since the animal owner is a breeder and does not want to treat the foal for business reasons. Situations like this emphasize that horses are not only companion animals, but also, as it were, commodities which can be used to generate financial profit and for business reasons. Decisions, based on economic factors, to avoid using medical options to treat the animal in order to avoid its suffering, and to protect the animal’s life, can distress veterinarians and lead to a sense of uncertainty.

The question of whether to euthanize a 21-year-old mare with unreachable owners and chronic bronchitis which have become refractory to treatment (F6) also resulted in a median value of 5. The cohort of respondents gave no clear statement as to whether they would euthanize the animal or not. Possible factors here include the fact that the level of suffering of the patient was not clearly stated, and the additional fact that a clear prior agreement and the informed consent of the owner were missing. Difficulties in obtaining client consent lead to challenging situations, especially as decisions can have negative consequences for veterinarians [31,32]. The Royal College of Veterinary Surgeons (RCVS) provides detailed guidance on communication and consent which states that veterinarians should provide first aid and act in cases of suffering animals even when the consent of the owner has not been obtained [33]. In Austria, veterinarians are legally required to provide first aid to an animal if the assistance is reasonable in view of the risk and does not violate other prevailing interests [34]. In this guidance, “first aid” refers to the immediate measures necessary to eliminate the risk to the animal’s life or to alleviate significant pain and suffering.

Although some items in the questionnaire resulted in significant differences in the tested variables, the effect sizes of the differences are very small. Against this background, it emerged that the veterinarians shared a similar mindset and had broadly homogenous attitudes to the issues addressed in the statements and case scenarios. This can be explained by the fact that most of the respondents probably acquired the same skills and knowledge in the course of their training at the University of Veterinary Medicine in Vienna, which has a monopoly on veterinary education in Austria. It is also significant that there is a rather small population of equine practitioners in Austria. As a result, Austrian veterinarians are a relatively close-knit group, and this probably has an effect on their attitude and mind-sets.

The limitation of the study might result from the fact that respondents were required to invest considerable time in the survey (56 items). Thus, it is likely that, especially, highly motivated veterinarians with a higher than average concern about euthanasia responded to the invitation to participate in the survey. Additionally, a number of respondents are not exclusively working as equine veterinarians and treating small animals or farm animals in addition to horses. These aspects may have created a bias. Further, a bias might result due to missing data, which limit the generalizability to the whole population of Austrian equine practitioners. 

## 5. Conclusions

Equine veterinarians are aware of the multifactorial influences on their decisions and consider several contextual and relational factors in the context of patient euthanasia. Respondents in the present survey gave clear indications as to how they weight animal- and owner-related factors, in addition to how these factors affect their decisions and help them to cope with euthanasia. They are aware of owners’ emotional bonds with their horses and financial background, and they reject requests for convenience euthanasia. Consideration of the animal’s prognosis, age, and previous life can make it easier for veterinarians to deal with euthanasia. Nevertheless, veterinarians are mindful of the importance of their informative and advisory role, which underlines the idea of shared decision-making processes in cases of euthanasia. Given that the present questionnaire-based survey was closely related to a similar survey in the field of small animal practice [14], further analysis examining small animal veterinarians and equine practitioners should be conducted to shed light on possible differences in the attitudes and beliefs of the two specialisms on the complex issue of euthanasia. This further investigation could clarify to what extent veterinarians are differently influenced by contextual factors due to their field of specialization and different working areas.

## Figures and Tables

**Table 1 animals-09-00044-t001:** Demographic and general variables (n = 64) presented for all survey respondents and separately for female and male respondents.

Demographic and General Variables	Total Study Population N = 64 (100%)	Female N = 43 (64%)	Male N = 21 (33%)	Missing Values
**Average age in years** (mean ± SD)	45 ± 9.5	40 ± 7.7	53 ± 6.4	-
**Average duration of professional activity in years** (mean ± SD)	16 ± 8.7	11.4 ± 6.4	24.4 ± 5.8	N = 2
**Work experience ≤ 15 years** (number (%))	28 (45.2)	27 (64.3)	1 (5.0)	N = 2
**Work experience > 15 years** (number (%))	34 (54.8)	15 (35.7)	19 (95.0)	N = 2
**60% or less working time with horses** (number (%))	28 (44.4)	18 (42.9)	10 (47.6)	N = 1
**Over 60% working time with horses** (number (%))	35 (55.6)	24 (57.1)	11 (52.4)	N = 1

**Table 2 animals-09-00044-t002:** Veterinarians’ level of agreement with 25 normative and descriptive statements in the context of euthanasia in equine practice (thematically grouped).

(1) General Statements on Euthanasia		Median (IQR)	Gender (Median (IQR))		Work Experience (Median (IQR))		Work Time with Horses (Median (IQR))	
*High Agreement*			Male	Female	≤15 years	>15 years	≤60%	>60%
**AI5** (n = 62)	Knowing that all veterinary medical, social, and economic options have been considered in the decision-making process makes it easier for me to deal with euthanasia.	9.0 (8.0;9.0)	9.0 (8.25;9.0)	9.0 (7.0;9.0)	9.0 (7.0;9.0)	9.0 (8.0;9.0)	9.0 (7.0;9.0)	9.0 (8.0;9.0)
**AII4** (n = 63)	It would be difficult for me to euthanize an animal against my convictions.	9.0 (9.0;9.0)	**9.0 *** (9.0;9.0)	**9.0 *** (9.0;9.0)	9.0 (9.0;9.0)	9.0 (9.0;9.0)	9.0 (9.0;9.0)	9.0 (9.0;9.0)
***Moderate Agreement***								
**AIII3** (n = 63)	I see euthanasia as an unavoidable evil of my professional responsibility.	8.0 (5.0;9.0)	9.0 (6.0;9.0)	7.5 (5.0;9.0)	8.0 (5;9)	8.0 (5.0;9.0)	8.0 (5,25;9.0)	8.0 (4.75;9.0)
**AIII7** (n = 64)	I see carefully considered euthanasia as a positive part of my practice as a veterinarian.	8.0 (6.0;9.0)	**9.0*** (7.5;9.0)	**7.0*** (5.0;9.0)	7.0 (6.0;9.0)	9.0 (5.5;9.0)	8.0 (5.25;9.0)	8.0 (6.0;9.0)
**AII8** (n = 64)	With increasing professional experience, it becomes easier for me to deal with the euthanasia of a horse.	7.0 (4.0;9.0)	8.0 (4.0;9.0)	7.0 (4.0;9.0)	7.0 (5.0;9.0)	7.0 (4.0;9.0)	8.0 (5.0;9.0)	7.0 (4.0;9.0)
***Ambivalent***								
**AI8** (n = 63)	Generally, it is still difficult for me to euthanize horses.	5.0 (2.0;8.5)	4.0 (2.0;6.5)	7.0 (2.75;9.0)	6.0 (2.0;9.0)	5.0 (2.0;9.0)	5.0 (2.0;7.0)	6.0 (3.0;9.0)
**(2) Patient-Centered Statements**								
***High Agreement***								
**AI2** (n = 64)	It is easier for me to deal with a medically indicated euthanasia if I know that the horse has only a short remaining lifespan.	9.0 (5.75;9.0)	9.0 (9.0;9.0)	8.0 (6.0;9.0)	8.0 (5.0;9.0)	9.0 (7.0;9.0)	9.0 (7.25;9.0)	8.0 (4.0;9.0)
**AII6** (n = 64)	Treating the dead horse in a respectful way is an important part of euthanasia.	9.0 (9.0;9.0)	9.0 (9.0;9.0)	9.0 (9.0;9.0)	9.0 (9.0;9.0)	9.0 (9.0;9.0)	9.0 (9.0;9.0)	9.0 (9.0;9.0)
***Moderate Agreement***								
**AIII2** (n = 63)	It is easier for me to deal with euthanasia if I know that I have done my best for the animal’s well-being.	8.0 (6.0;9.0)	8.0 (5.5;9.0)	7.0 (5.75;9)	8.0 (7.0;9.0)	7.0 (5.0;8.0)	7.5 (5.25;9.0)	8.0 (6.0;9.0)
**AII3** (n = 64)	It is easier for me to deal with euthanasia if I have the impression that the animal has lived a rich life until its death.	7.0 (5.0;9.0)	6.0 (3.5;8.0)	7.0 (5.0;9.0)	8.0 (5.5;9.0)	7.0 (5.0;8.0)	7.0 (5.0;9.0)	7.0 (5.0;9.0)
**AIII5** (n = 64)	The animal’s advanced (high) age makes it easier for me to deal with euthanasia.	7.0 (5.0;8.0)	8.0 (4.5;9.0)	7.0 (5.0;8.0)	7.0 (3.25;8.0)	7.0 (5.0;8.0)	**8.0*** (7.0;8.0)	**7.0*** (3.0;8.0)
***Ambivalent***								
**AI1** (n = 64)	Knowing that I have done my best for the horse’s well-being makes it easier for me to deal with the horse’s suffering if the owner does not agree to euthanasia.	5.0 (2.0;8.0)	5.0 (3.5;8.5)	5.0 (2.0;8.0)	6.0 (2.0;8.0)	5.0 (2.5;8.0)	7.0 (2.25;9.0)	4.0 (2.0;7.0)
**(3) Client-Centered Statements**		**Median** **(IQR)**	**Gender** **(Median (IQR))**		**Work Experience** **(Median (IQR))**		**Work Time with Horses** **(Median (IQR))**	
***High Agreement***			**Male**	**Female**	**≤15** **years**	**>15** **years**	**≤60%**	**>60%**
**AI3** (n = 63)	If the horse owner is thoroughly informed I find it easier to deal with euthanasia.	9.0 (7.0;9.0)	9.0 (7.0;9.0)	9.0 (7.0;9.0)	9.0 (8.0;9.0)	9.0 (7.0;9.0)	**8,5 *** (7.0;9.0)	**9.0*** (7.75;9.0)
**AI4** (n = 63)	The owner’s understanding that the euthanasia of his/her horse is necessary makes it easier for me to deal with euthanasia.	9.0 (8.0;9.0)	9.0 (7.25;9.0)	9.0 (8.0;9.0)	9.0 (8.0;9.0)	9.0 (8.0;9.0)	9.0 (8.0;9.0)	9.0 (8.0;9.0)
**AII7** (n = 64)	Treating the owners with compassion is a central part of euthanasia.	9.0 (9.0;9.0)	9.0 (8.0;9.0)	9.0 (9.0;9.0)	**9.0*** (8.0;9.9)	**9.0*** (9.0;9.0)	9.0 (8.0;9.0)	9.0 (9.0;9.0)
***Ambivalent***								
**AII2** **(n = 64)**	If the patient’s owner is thoroughly informed I find it easier to deal with the suffering of the animal, even if the owners decide against euthanasia of the horse.	5.0 (2.0;6.25)	5.0 (2.0;9.0)	5.0 (1.0;6.0)	5.0 (2.0;8.0)	5.0 (2.0;6.0)	5.0 (2.25;6.0)	5.0 (2.0;7.0)
***Disagreement***								
**AI7** (n = 64)	It is easier for me to deal with euthanasia if the owners are present during the procedure.	3.5 (1.0;5.0)	5.0 (1.0;6.0)	3.0 (1.0;5.0)	3.0 (1.0;5.0)	4.0 (2.0;5.0)	4.0 (2.0;5.0)	3.0 (1.0;7.0)
**AI6** (n = 63)	It is easier for me to euthanize a horse if I have the impression that the animal’s owners have no close relationships with the animal.	3.0 (1.0;6.0)	3.0 (1.0;6.5)	2.5 (1.0;6.0)	1.0 (1.0;7.0)	3.0 (1.5;5.0)	3.5 (1.0;6.75)	2.0 (1.0;5.25)
**AII5** (n = 63)	The presence of the horse owner during euthanasia tends to cause more problems.	3.0 (2.0;7.0)	5.0 (2.0;7.5)	3.0 (1.0;7.0)	3.0 (2.0;7.0)	4.0 (1.5;7.0)	4.5 (2.0;7.75)	2.5 (1.0;5.25)
**AIII4** (n = 63)	Knowing that my influence on the owner’s decision is limited makes it easier for me to deal with euthanasia.	3.0 (1.0;5.0)	5.0 (2.0;6.0)	2.0 (1.0;5.0)	3.0 (1.25;5.0)	2.0 (1.0;5.0)	2.5 (2.0;5.75)	2.5 (1.0;5.0)
***Strong Disagreement***								
**AIII1** (n = 64)	I would euthanize an animal in cases without medical indication for euthanasia if I am afraid that the owner will neglect the horse.	2.0 (1.0;5.0)	2.0 (1.0;5.0)	1.0 (1.0;3.0)	1.5 (1.0;5.0)	2.0 (1.0;5.0)	2.0 (1.0;4.75)	1.0 (1.0;5.0)
**AIII6** (n = 64)	I would euthanize an animal in cases without medical indication for euthanasia if I am afraid that the owner will see another veterinarian.	1.0 (1.0;1.0)	1.0 (1.0;1.5)	1.0 (1.0;1.0)	1.0 (1.0;1.0)	1.0 (1.0;1.5)	1.0 (1.0;1.0)	1.0 (1.0;1.0)
**(4) Technical and Medical Statements**		**Median** **(IQR)**	**Gender** **(Median (IQR))**		**Work Experience** **(Median (IQR))**		**Work Time with Horses** **(Median (IQR))**	
***High Agreement***			**Male**	**Female**	**≤15** **years**	**>15** **years**	**≤60%**	**>60%**
**AI10** (n = 64)	A technically flawless and uncomplicated euthanasia of the animal makes it easier for me to deal with euthanasia.	9.0 (9.0;9.0)	9.0 (9.0;9.0)	9.0 (8.0;9.0)	9.0 (9.0;9.0)	9.0 (8.5;9.0)	9.0 (9.0;9.0)	9.0 (8.0;9.0)
***Moderate Agreement***								
**AII1** (n = 64)	Effective pain therapy makes it easier for me to deal with a horse’s suffering.	8.0 (7.75;9.0)	9.0 (8.0;9.0)	8.0 (7.0;9.0)	8.0 (7.25;9.0)	8.0 (7.0;9.0)	8.0 (7.0;9.0)	9.0 (8.0;9.0)
**AI9** (n = 64)	Careful planning and the right moment make it easier for me to deal with euthanasia.	6.0 (3.75;6.0)	7.0 (4.0;9.0)	6.0 (3.0;9.0)	7.0 (4.25;9.0)	5.0 (3.0;9.0)	7.0 (5.0;9.0)	6.0 (2.0;9.0)

For the statements, the veterinarians had the possibility to gauge their agreement with the statement from 1 = ”I do not agree at all” to 9 = ”I completely agree”. * Significant difference if *p* < 0.05 and are based on a Mann-Whitney U test.

**Table 3 animals-09-00044-t003:** Veterinarians’ level of agreement with the endorsement of euthanasia in ten scenarios in equine practice.

“Convenience Euthanasia” Scenarios with External Pressure for Euthanasia		Median (IQR)	Gender (Median (IQR))		Work Experience (Median (IQR))		Work Time with Horses (Median (IQR))	
			Male	Female	≤15 years	>15 years	≤60%	>60%
**F2** (n = 64)	A 15-year-old mare with a fresh wound with involvement of the left front fetlock joint is presented to you. You give the horse a good prognosis for future use as a leisure riding horse. The animal owner refuses therapy for financial reasons and wants you to euthanize the animal.	1.0 (1.0;4.0)	2.0 (1.0;5.0)	1.0 (1.0;4.0)	1.0 (1.0;3.75)	2.0 (1.0;5.5)	2.0 (1.0;4.75)	1.0 (1.0;4.0)
**F3** (n = 64)	An owner of two Icelandic horses dies and bequeaths her two animals to a friend for further care. The horses have been kept together since they were foals, for more than 30 years. One of the animals dies at the age of 37. In the will of the deceased owner, the following is stipulated: In the event of the death of one of the two horses, the other is to be euthanized in order to save the surviving horse from suffering through separation. The surviving horse is in excellent condition for its age and there is no reason why it should not live for a few more years. The new owner asks you to euthanize the surviving horse.	1.0 (1.0;3.25)	1.0 (1.0;1.5)	1.0 (1.0;5.0)	1.0 (1.0;2.5)	1.0 (1.0;4.25)	**1.5 *** **(1.0;5.0)**	**1.0 *** **(1.0;1.0)**
**F4** (n = 64)	The owner of a young stallion contacts you. The stallion has a complicated contusion laceration of the left hind leg. You give the horse a very good prognosis for full recovery. The animal owner refuses therapy because he is concerned that the stallion could not cope with stall confinement for the required 12-week period. He believes it would affect the horse’s quality of life to the point that he would like you to euthanize the stallion.	1.0 (1.0;1.0)	1.0 (1.0;1.5)	1.0 (1.0;1.0)	**1.0 *** **(1.0;1.0)**	**1.0 *** **(1.0;1.25)**	1.0 (1.0;1.0)	1.0 (1.0;1.0)
**F5** (n = 64)	You are called to a mare with a foal. The foal shows symptoms of severe colic. The owner does not want to have the proposed operation carried out, as the costs would exceed the foal’s monetary value, and wants you to euthanize the foal.	5.0 (4.0;8.0)	7.0 (4.5;8.5)	5.0 (3.0;8.0)	5.0 (3.25;9.0)	5.0 (4.0;8.0)	5.5 (3.0;8.0)	5.0 (4.0;8.0)
**F7** (n = 64)	A horse owner asks you to euthanize her 25-year-old gelding because she intends to travel with her family for an extended period of time and does not want to uproot the horse at his age by moving him to a boarding stable.	1.0 (1.0;2.0)	1.0 (1.0;2.0)	1.0 (1.0;2.0)	1.0 (1.0;1.75)	1.0 (1.0;2.0)	1.0 (1.0;2.75)	1.0 (1.0;1.0)
**“Owners refusal to euthanize” Scenario with External Pressure Against Euthanasia**								
**F1** (n = 64)	An owner turns to you with his severely laminitic mare. You know that he is very attached to his horse and does not want to part with it. From your point of view, euthanasia is clearly indicated. After a detailed explanation, the owner still refuses euthanasia.	2.0 (1.0;5.25)	2.0 (1.0;6.0)	1.0 (1.0;2.0)	2.0 (1.0;7.5)	1.5 (1.0;3.5)	2.0 (1.0;5.0)	2.0 (1.0;8.0)
**“Notification” Inform Veterinary Officer**		**Median** **(IQR)**	**Gender** **(Median (IQR))**		**Work experience** **(Median (IQR))**		**Work Time with Horses** **(Median (IQR))**	
			**Male**	**Female**	**≤15** **years**	**>15** **years**	**≤60%**	**>60%**
**F8** (n = 63)	You are presented with a 25-year-old mare which appears to be severely neglected, is already highly cachectic and shows no interest in her surroundings. Due to the poor general condition of the animal, you believe that it is already close to death and the prognosis is infaust. Do you think that the official veterinarian should be informed in this case?	9.0 (7.0;9.0)	8.0 (6.0;9.0)	9.0 (7.0;9.0)	9.0 (8.0;9.0)	9.0 (7.0;9.0)	9.0 (7.0;9.0)	9.0 (8.0;9.0)
**“Responsibility” Decision Making Process**								
**F6** (n = 64)	You receive a phone call from a woman who looks after the 21-year-old mare of friends who just left for a four-week trekking tour and are not reachable. You have been treating the horse for six months for chronic bronchitis. The horse initially responded well to therapy but is now deteriorating and does not respond to treatment anymore. The woman refuses to make a decision regarding euthanasia and cannot tell you what the owners might want.	5.0 (2.75;8.0)	5.0 (2.5;8.5)	5.0 (2.0;7.0)	5.0 (2.0;6.75)	6.0 (2.75;8.25)	6.0 (3.0;8.75)	4.0 (2.0;7.0)
**F9** (n = 64)	You are presented with a 12-year-old warmblood gelding with a closed fracture of the proximal phalanx. You give the horse a 50% chance of survival with surgical intervention. The owners are indecisive and ask: "What would you do if it was your animal?" Would you make a personal recommendation and thus possibly influence and effect make the decision?	7.0 (5.0;9.0)	7.0 (5.0;9.0)	7.0 (5.0;9.0)	7.0 (5.0;8.75)	8.0 (5.0;9.0)	8.0 (6.25;9.0)	7.0 (5.0;9.0)
**“Technical aspect” Use of Sedation**								
**F10** (n = 64)	You are called to a Shetland pony in lateral recumbency which was hit by a car. The pony is apathetic. During the examination, you determine that the pony has sustained fatal injuries and recommend euthanasia, which the owners consent to. Would you sedate the pony before administering the euthanasia medication despite its apathetic state?	1.0 (1.0;1.0)	1.0 (1.0;1,5)	1.0 (1.0;1.0)	1.0 (1.0;1.75)	1.0 (1.0;1.0)	1.0 (1.0;1.0)	1.0 (1.0;3.0)

For the scenarios F1–F7, the veterinarians were asked “What is your attitude towards euthanasia?” and had the possibility to gauge their agreement with euthanasia in this case from 1 = ”I reject euthanasia” to 9 = ”I fully agree with euthanasia”. In scenario F8 the question was about the necessity to notify an official veterinarian, with the answer options ranging from 1 = ”rejection” to 9 = ”agreement”. The answer options for scenario F9, asking about the willingness to take decision concerning euthanasia in the place of the owner, ranged from 1 = ”I am sure I would make no recommendation” to 9 = ”I am sure I would make a recommendation”. Scenario F10 includes a technical aspect and veterinarians had the possibility to choose from 1 = ”I am sure I would sedate the animal” to 9 =”I am sure I would not sedate the animal”. * Significant difference if *p* < 0.05 and are based on Mann-Whitney U test.

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
