# Peer review of "Austrian Veterinarians’ Attitudes to Euthanasia in Equine Practice"

_animals, 2019, doi:10.3390/ani9020044_

Round 1

Reviewer 1 Report

This is an important paper to get into the literature. I can envision it being an excellent resource for Veterinary School ethics classes and similar. It is well written and has been nicely edited.  (There are a few minor questions/suggestions/edits on the attached , edited PDF.

Author Response

Thank you very much for your positive comments as well as constructive suggestions for improvement of our manuscript. All revisions and addressed questions of Reviewer 1 can be found in the attached file. 

Reviewer 2 Report

The present paper report an anonymous questionnaire-based survey of Austrian equine veterinarians’ examines the attitudes to the euthanasia of equine patients in a range of scenarios, and to identify factors which may influence decisions on the ending of a horse’s life. 

The work appears sufficiently well conducted and the results reported are pertinent.

The objectives of the review are of interest and fit well within the scope of the journal.

I suggest the authors to add the following reference (line 48):

Passantino A., Fenga C., Morciano C., Morelli C., Russo M., Di Pietro C., Passantino M. (2006). Euthanasia of companion animals: a legal and ethical analysis. Annali dell’Istituto Superiore di Sanità, 42(4) 491-495.

In my opinion, the manuscript could be accepted for publication in Animals with the above-mentioned suggestions.

Author Response

Thank your very much for your valuable review. We included the suggested reference in our revised version of the manuscript [line 48].

Reviewer 3 Report

Extremely important topic and findings. Thank you for conducting this study on a challenging topic we all face. Being a responsible equine owner or veterinarian includes assessing quality of life and making such decisions. Interesting to see the different responses based on age, gender, types of practice. Additional time should be spent in vet school addressing empathy. 

Author Response

Thank your very much for your positive comments on our manuscripts.